# Influence of Forging and Heat Treatment on the Microstructure and Mechanical Properties of a Heavily Alloyed Ingot-Metallurgy Nickel-Based Superalloy

**Valery Imayev [1], Shamil Mukhtarov [1,*], Kamilla Mukhtarova [2], Artem Ganeev [1], Ruslan Shakhov [1], Nikolay Parkhimovich [1] and Aleksander Logunov [3]**

[1]  Institute for Metals Superplasticity Problems of Russian Academy of Sciences, 39 Stepan Khalturin, 450001 Ufa, Russia; vimayev@imsp.ru (V.I.); artem@imsp.ru (A.G.); shakhov-rv@yandex.ru (R.S.); pn@imsp.ru (N.P.)

[2]  Faculty of Science, Eötvös Loránd University, 1/A 2 Pázmány Péter sétány, 1117 Budapest, Hungary; kamillamsh@student.elte.hu

[3]  PJSC "UEC-Saturn", 163 Lenin Ave., 152903 Rybinsk, Russia; ilya.hryaschev@uec-saturn.ru

*  Correspondence: shamil@anrb.ru; Tel.: +7-347-282-3819

**Abstract:** The newly designed ingot-metallurgy nickel-based superalloy SDZhS-15 intended for disc applications at operating temperatures up to 800–850 °C was subjected to homogenization annealing and canned forging at subsolvus temperatures, followed by solid solution treatment and ageing. Mostly a fine-grained recrystallized microstructure was obtained in the forgings. It was revealed that post-forging solid solution treatment at $T > (T_s\text{-}50)$, where $T_s$ is the $\gamma'$ solvus temperature, led to a significant $\gamma$ grain growth, which in turn led to a decrease in strength and ductility of the superalloy. The solution treatment at $(T_s\text{-}60)$–$(T_s\text{-}50)$ allowed to save fine $\gamma$ grains ($d_\gamma = 10$–$20$ μm) and to provide the formation of secondary $\gamma'$ precipitates with a size of around 0.1 μm. In the forged and heat-treated conditions, the superalloy demonstrated superior mechanical properties, particularly excellent creep resistance at 650–850 °C in the stress range of 400–1200 MPa. Microstructure examination of the creep-tested samples showed that a decrease in the creep resistance at 850 °C can be associated with enhanced diffusivity along $\gamma$ grain and $\gamma/\gamma'$ interphase boundaries leading to formation of cracks along the boundaries. In spite of the heavy alloying, the topologically close-packed phases were not detected in the superalloy, including in the creep tested samples.

**Keywords:** nickel-based superalloy; microstructure; forging; solid solution treatment; ageing; mechanical properties; creep resistance

---

## 1. Introduction

Improving the energy efficiency of gas turbine engines (GTE) and similar energy-conversion systems is dependent on the development of new advanced materials with enhanced strength, heat resistance, and/or reduced specific weight. In particular, increasing demands are being placed on nickel-based superalloys, which are widely used for manufacturing of rotary engine parts such as high pressure discs in GTE [1]. For this purpose, novel heavily alloyed disc superalloys with a higher content of the $\gamma'$(Ni$_3$Al) forming elements, such as Russian superalloy SDZhS-15, are being designed. In this superalloy, the content of (Al + Ti + Nb + Ta) is 14.8 at. % and the ratio (Ti + Nb + Ta)/Al (in at. %) is 0.86, which indicates a significant solid solution hardening of the $\gamma'$ phase. The $\gamma$ matrix chemistry was optimized through a careful balance between the solid solution elements, such as Cr, Co, Mo, W, and Re [2,3]. Generally, the goals of alloying were to: (i) increase the $\gamma'$ solvus temperature

up to 1200 °C and higher; (ii) increase the solid solution hardening of the $\gamma$ matrix and the $\gamma'$ phase; (iii) maintain oxidation and corrosion resistances via significant alloying with Cr; and (iv) avoid the precipitation of undesirable topologically close-packed (TCP) phases. Long-term homogenization heat treatment and hot forging processing have been developed for the superalloy that allowed obtaining forgings with a homogeneous recrystallized fine-grained microstructure [2,3]. It is worth noting that the superalloy showed quite reasonable mechanical properties in comparison with those of other disc nickel-based superalloys even in the fine-grained condition not subjected to solid solution treatment [2]. Nevertheless, to have properly balanced mechanical properties, particularly higher creep resistance, the fine-grained condition of the SDZhS-15 superalloy should be subjected to post-forging solid solution treatment and ageing to obtain a high volume fraction of $\gamma'$ precipitates [4–6]. In so doing, the $\gamma$ grain growth should be apparently avoided to retain a small $\gamma$ grain size. The refined microstructure ($d_\gamma$ = 10–20 μm) is particularly important for the hub part of the GTE discs [4,5,7], which is subjected to high tensile, fatigue, and thermocyclic stresses under service conditions.

The present work aimed to study the effect of post-forging heat treatment on the microstructure and mechanical properties of the SDZhS-15 superalloy. The heat treatment included different solid solution treatment followed by air cooling with a certain cooling rate, and ageing. Tensile properties and creep resistance of the superalloy were evaluated in different microstructural conditions.

## 2. Materials and Methods

### 2.1. Initial Material

The SDZhS-15 superalloy with a nominal chemical composition of Ni-29.5(Cr,Co)-14.8 (Al,Ti,Nb,Ta)-3.8(Mo,W,Re)-0.5(C,La,Y,Ce,B) (in at.%) (Ni-28(Cr,Co)-12.5(Al,Ti,Nb,Ta)-9(Mo,W,Re)-0.17(C,La,Y,Ce,B) (in wt.%) was manufactured by vacuum induction melting as ingots with a size of about Ø127 mm × 400 mm. The exact composition of the superalloy is not indicated for confidentiality reasons. The superalloy composition was found to be very close to its nominal composition. The $\gamma'$ (Ni$_3$Al) solvus temperature was determined as $T_s$ = 1220 ± 5 °C via quenching experiments [2].

### 2.2. Processing Methods

Several workpieces were cut from the as-cast superalloy, which were subjected to different processing routes. Various microstructural conditions were produced. The as-cast material was subjected to long-term homogenization annealing at near the $\gamma'$ solvus temperature and solid solution treatment at 1210 °C, followed by air cooling and two-stage ageing at 860 °C (6 h) and 750 °C (32 h). The produced microstructure was designated as condition 1. The basic processing routes included long-term homogenization annealing at near the $\gamma'$ solvus temperature followed by slow cooling, and canned forging with intermediate annealing at subsolvus temperatures, followed by air cooling and decanning as described elsewhere [2,3]. Using this technique, sound fine-grained forgings with an approximate size of Ø100 mm × 15 mm were obtained. The post-forging heat treatment included either two-stage ageing at 860 and 750 °C (condition 2) or solid solution treatment during 1 h at 1160, 1170, 1180, 1190, and 1200 °C, followed by air cooling and two-stage ageing at 860 and 750 °C. The air cooling was conducted with a rate of about 10 °C/s. The microstructural conditions obtained via solution treatment at 1170 and 1200 °C were designated in the text as conditions 3 and 4, respectively. Table 1 represents the processing routes used in the present work.

**Table 1.** Processing of the SDZhS-15 superalloy.

| Processing Route | Designation in the Text |
|---|---|
| Cast + HA [1] + ST [2] (1210 °C) + A [3] | Condition 1 |
| Cast + HA + HF [4] + A | Condition 2 |
| Cast + HA + HF + ST (1160 °C) + A | - |
| Cast + HA + HF + ST (1170 °C) + A | Condition 3 |

**Table 1.** *Cont.*

| Processing Route | Designation in the Text |
|---|---|
| Cast + HA + HF + ST (1180 °C) + A | - |
| Cast + HA + HF + ST (1190 °C) + A | - |
| Cast + HA + HF + ST (1200 °C) + A | Condition 4 |

[1] HA—homogenization annealing, [2] ST—solid solution treatment, [3] A—ageing, [4] HF—hot forging with intermediate annealing.

### 2.3. Microstructural Examination

The microstructure examination was carried out using scanning electron microscopy (SEM). SEM in the secondary electron (SE) and back-scattering electron (BSE) mode was performed in a Mira-3 Tescan microscope (TESCAN, Brno-Kohoutovice, Czech Republic), which was equipped with an energy-dispersive X-ray analysis system. Energy-dispersive X-ray spectroscopic (EDS) analysis was carried out for the cast condition subjected to homogenization annealing followed by slow cooling with a rate of 25 °C/h. The slow cooling led to coarsening of the $\gamma'$ phase and significant reduction of the fraction of the dispersed $\gamma'$ precipitates. That allowed us to evaluate the partitioning behavior of the alloying elements taking into account $\gamma$ grain areas free of the $\gamma'$ precipitates versus coarse $\gamma'$ particles. Not less than 50 measurements were made in total for $\gamma$ grains and $\gamma'$ particles. The mean partition coefficients ($k_{\gamma/\gamma'}$) were calculated for the alloying elements. The electron backscatter diffraction (EBSD) analysis was performed with a scan-step size of 0.5 μm using the HKL Channel 5 software (Oxford Instruments HKL, Hobro, Denmark). Due to the experimental error in evaluating orientations by EBSD [8], the grain/interphase boundaries having misorientation angle less than 2° were excluded from the data analysis. The grain boundaries having misorientation angle more than 15° were assumed as high-angle ones.

### 2.4. Mechanical Tests

The processed workpieces in conditions 1–4 were used to prepare the samples for tensile and creep testing. The tensile samples had a gauge section of $10 \times 3 \times 2$ mm$^3$. The tensile tests were performed at $T = 20, 650, 750,$ and 850 °C. A constant speed of the testing machine was 0.5 mm/min that corresponded to the initial strain rate of $\dot{\varepsilon} = 8.3 \times 10^{-4}$ s$^{-1}$. Three samples per point were tensile tested. The creep samples had a gage section of $17 \times 3 \times 3$ mm$^3$. The creep tests were carried out at $T = 650, 750, 800,$ and 850 °C. The two-arm load machine having a lever arm ratio of 20:1 was used for the creep tests. The samples were cut so that the flat surfaces of the samples were parallel to the forging direction. Before testing, the samples were mechanically ground. All mechanical tests were carried out in air. A Larson-Miller parameter (*LMP*) relationship to predict creep rupture life for the superalloy was calculated by Equation (1) [9–11]:

$$LMP = T \times [\log t_{rup} + C_{LM}], \tag{1}$$

where $T$ is the creep testing temperature in $K$, $t_{rup}$ is the creep rupture life in hours, and $C_{LM}$ is the Larson-Miller constant, which was chosen as 20 [10,11].

## 3. Results and Discussion

### 3.1. Microstructure Characterization

Figure 1 shows the microstructure of the superalloy in the cast and heat treated condition (condition 1). Dendritic segregation observed in the as-cast ingot was dissolved during the homogenization annealing [2]. The $\gamma$ grain size was in the range of 100–500 μm (Figure 1a). The primary $\gamma'$ phase was not observed. The mean size of the secondary $\gamma'$ precipitates was $d_{\gamma'} = 0.15 \pm 0.02$ μm (Figure 1b). The volume fraction of the secondary $\gamma'$ precipitates was defined as 68%. The secondary

$\gamma'$ phase precipitated during air cooling after solution treatment and ageing. In the microstructure there were carbides having a complex composition (Me$_x$C$_y$). Their volume fraction was found to be about 2%. Unfavorable TCP phases were not detected in condition 1.

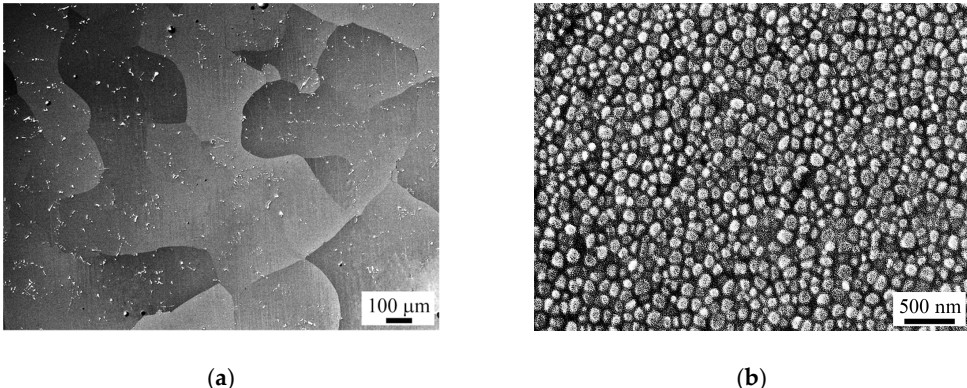

(a)  (b)

**Figure 1.** The back-scattering electron (BSE) images of the cast and heat treated superalloy (condition 1): (**a**) coarse $\gamma$ grains and carbides (white particles); (**b**) $\gamma'$ precipitates.

Figure 2a,b represents the SE image and the EBSD orientation map obtained from the central part of the forged workpiece. Forging with intermediate annealing at subsolvus temperatures resulted in mostly recrystallized and fine-grained microstructure. There were also individual coarse $\gamma$ grains with a size of 30–100 μm. As a rule, the coarse $\gamma$ grains contained low-angle boundaries (Figure 2b). The mean size of recrystallized $\gamma$ grains was defined as $d_\gamma = 12 \pm 1.5$ μm. The fraction of high-angle grain/interphase boundaries was defined as 80%.

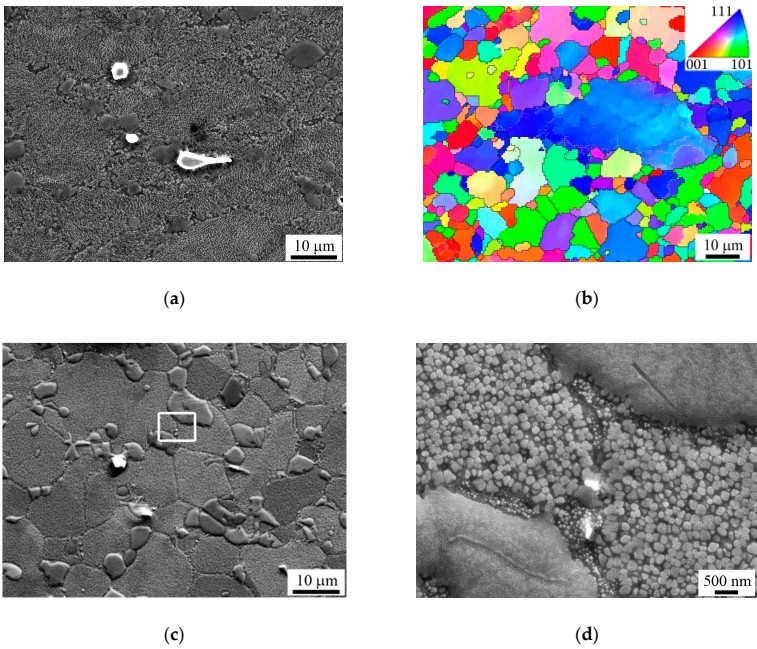

(a)  (b)

(c)  (d)

**Figure 2.** The microstructure images of the superalloy after hot forging with intermediate annealing (**a**,**b**) and ageing (condition 2) (**c**,**d**): (**a**) secondary electron (SE) image showing the primary, secondary $\gamma'$ phase and white carbides, (**b**) normal-direction electron backscatter diffraction (EBSD) (inverse-pole-figure) map, the forging axis is vertical, high- and low-angle grain/interphase boundaries are indicated by black and white lines, respectively; (**c**,**d**) SE images illustrating the primary, secondary, tertiary $\gamma'$ phase and white carbides including fine carbide particles (**d**) precipitated during ageing. The image (**d**) was obtained from the area marked by the rectangle in (**c**).

Primary $\gamma'$ precipitates not dissolved at forging temperatures had a size of 2–10 μm and occupied about 15 vol.%. They were mostly located along $\gamma$ grain boundaries (Figure 2a,c). The secondary and tertiary $\gamma'$ precipitates were formed during air cooling from the forging temperature and ageing. Their average size was $d_{\gamma'}$ = 0.15 ± 0.02 μm and $d_{\gamma'}$ = 0.05 ± 0.01 μm, respectively (Figure 2d). The tertiary $\gamma'$ precipitates were observed along the $\gamma$ grain boundaries and their volume fraction was not more than 1–1.5%. The volume fraction of the secondary and tertiary $\gamma'$ phase was about 53%. Small carbide particles having a size of less than 1 μm were precipitated during ageing and observed along the grain/interphase boundaries (Figure 2c,d). The carbide particles occupied approximately the same volume fraction as in condition 1. Unfavorable TCP phases were not detected in condition 2, which is consistent with the earlier performed work [2].

Figure 3 shows the BSE and SE images obtained after hot forging with intermediate annealing, followed by solid solution treatment at 1170 and 1200 °C and ageing. Figure 4 shows the dependencies of the $\gamma$ grain size and the volume fraction of the primary $\gamma'$ phase obtained after solid solution treatment and ageing. After solution treatment at 1160 °C and ageing, the mean $\gamma$ grain size was not appreciably changed as compared with the forged condition and averaged $d_{\gamma}$ = 14 ± 1.5 μm. The increase in the solution treatment temperature led to $\gamma$ grain growth due to static recrystallization and dissolution of the $\gamma'$ phase, which restricted the $\gamma$ grain growth. As the solution treatment temperature increased from 1160 to 1200 °C, the $\gamma$ grain size increased from 14 to about 60 μm and the volume fraction of the primary $\gamma'$ phase decreased from 15% to 4.5% (Figure 3a,c and Figure 4a,b). The mean size of the secondary $\gamma'$ precipitates after solution treatment at 1160–1200 °C and ageing was varied within $d_{\gamma'}$ = 0.09–0.13 μm (Figure 3b,d and Figure 4c). The tertiary $\gamma'$ precipitates were sometimes observed along $\gamma$ grain boundaries, their size was smaller than 0.1 μm and the volume fraction was always not more than 1% (Figure 3b). Most probably, they were formed during ageing. The volume fractions of the secondary and tertiary $\gamma'$ phase in conditions 3 and 4 were determined as 58 and 63.5%, respectively. The volume fraction of carbides after solution heat treatment and ageing (conditions 3 and 4) was not noticeably changed as compared with conditions 1 and 2. TCP phases were not detected in conditions 3 and 4.

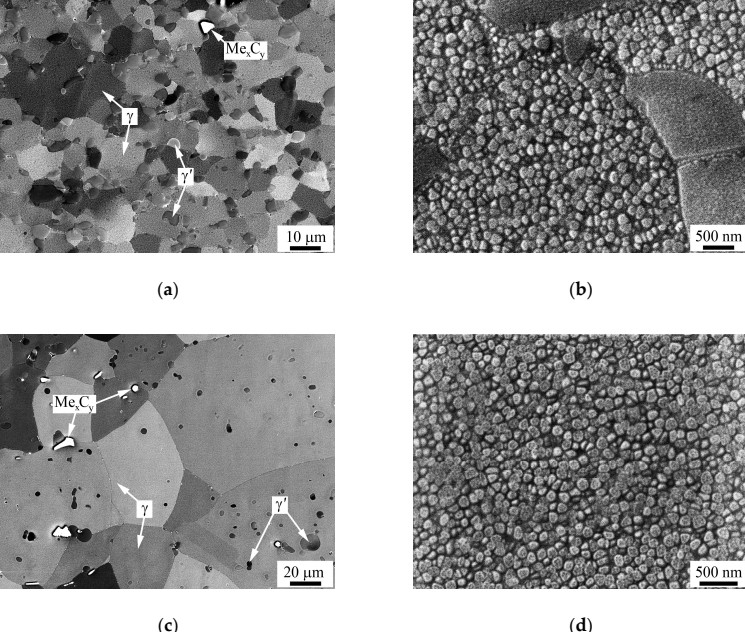

(a)

(b)

(c)

(d)

**Figure 3.** The microstructure images of the superalloy after hot forging, solid solution treatment at different temperatures and ageing: (**a**,**b**) condition 3, (**c**,**d**) condition 4. (**a**,**c**) The arrows show the $\gamma$ grains, primary $\gamma'$ phase, and carbides (BSE), (**b**,**d**) $\gamma'$ precipitates (SE).

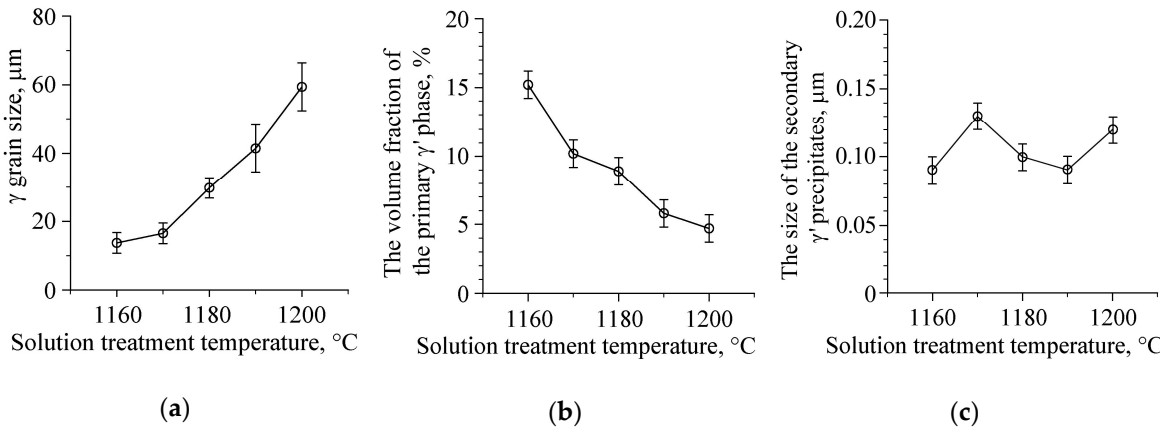

**Figure 4.** The dependencies of (**a**) the $\gamma$ grain size, (**b**) the volume fraction of the primary $\gamma'$ phase and (**c**) the size of the secondary $\gamma'$ precipitates obtained after solid solution treatment at different temperatures followed by ageing.

As mentioned in the introduction, the optimal $\gamma$ grain size lies in the range of 10–20 µm. Therefore, the temperature range 1160–1170 °C is the most appropriate for solid solution treatment. After solution treatment in this temperature range, the $\gamma$ grain size was defined as $d_\gamma$ = 14–16 µm, the volume fraction of the primary $\gamma'$ phase was relatively low (10–15%), and the mean size of the secondary $\gamma'$ precipitates was defined as $d_{\gamma'}$ = 0.09–0.13 µm. Thus, the post-forging solid solution treatment at 1160–1170 °C resulted in formation of dispersed $\gamma'$ precipitates, while retaining refined $\gamma$ grains which is a good prerequisite for achieving enhanced mechanical properties.

### 3.2. Partitioning Behavior of The Alloying Elements

The cast superalloy subjected to homogenization annealing followed by slow cooling was used to evaluate the partitioning behaviors of the alloying elements. As mentioned, the slow cooling provided coarsening of the $\gamma'$ phase and significant reduction of the fraction of the $\gamma'$ precipitates. EDS analyses were done taking into consideration $\gamma$ grains versus coarse $\gamma'$ particles.

Table 2 represents the calculated quantitative results averaged over many spectrums. One can see that the partitioning preference is $\gamma > \gamma'$ for Cr, Co, Mo, W, and Re, and $\gamma' > \gamma$ for Al, Ti, Nb, Ta, and Ni. One should expect that the highest solid solution hardening (per unit of concentration of the alloying element) in the $\gamma$ and $\gamma'$ phase will correspond to the highest and the lowest partition coefficient $k_{\gamma/\gamma'}$, respectively. Therefore, the highest solid solution hardening (per unit of concentration) in the $\gamma$ and $\gamma'$ phase is reached by alloying with Re, Mo and Ti, Ta, respectively. Note that the highest partition coefficient $k_{\gamma/\gamma'}$ obtained for rhenium explains why this alloying element is among the most effective for solid solution hardening of the $\gamma$ phase.

**Table 2.** The averaged partition coefficients $k_{\gamma/\gamma'}$ obtained for the alloying elements of the superalloy.

| The Partition Coefficient | Alloying Elements | | | | | | | | | |
|---|---|---|---|---|---|---|---|---|---|---|
| | Al | Ti | Cr | Co | Ni | Nb | Mo | Ta | W | Re |
| $k_{\gamma/\gamma'}$ | 0.32 | 0.24 | 5.18 | 1.76 | 0.68 | 0.39 | 4.37 | 0.30 | 1.87 | 7.92 |

### 3.3. Tensile Properties

Figure 5 represents the tensile properties of the superalloy at 20, 650, and 750 °C obtained in the conditions 1–4. The forged and heat-treated conditions 2–4 demonstrated appreciably higher strength (ultimate tensile strength, UTS) and ductility as compared with those of the cast and heat treated condition 1. This should be ascribed to finer $\gamma$ grain size in conditions 2–4. The decrease of the strength properties (UTS and yield strength (YS)) in conditions 3 and 4 in contrast to those in condition

2 is associated with coarser $\gamma$ grain size in conditions 3 and 4. Conditions 3 and 4 showed similar tensile properties at 650 and 750 °C, although the $\gamma$ grain size in condition 4 was coarser than that in condition 3 (60 μm vs. 16 μm). Apparently, the higher volume fraction of the secondary $\gamma'$ precipitates in condition 4 in contrast to condition 3 (63.5% vs. 58%) promoted higher strength that compensated for the loss of strength due to larger $\gamma$ grain size in condition 4. Comparing conditions 1, 3, and 4, one can see that the yield strength values in these conditions were similar (especially in conditions 1 and 4). This suggests that the yield strength value weakly depends on the $\gamma$ grain size in the range of $d\sim$20–200 μm and mostly depends on the volume fraction of the secondary $\gamma'$ precipitates. Note that the higher volume fraction of the secondary $\gamma'$ precipitates in conditions 1 and 4 as compared with that in condition 3 (68 and 63.5% vs. 58%) apparently compensated for the loss of the yield strength due to larger $\gamma$ grain size in conditions 1 and 4. At the same time, the UTS values and the elongations in conditions 3 and 4 were found to be appreciably higher than those in condition 1. This should be attributed to finer $\gamma$ grain size in conditions 3 and 4, which promoted a uniform development of plastic deformation that in turn promoted attaining higher elongations and UTS values.

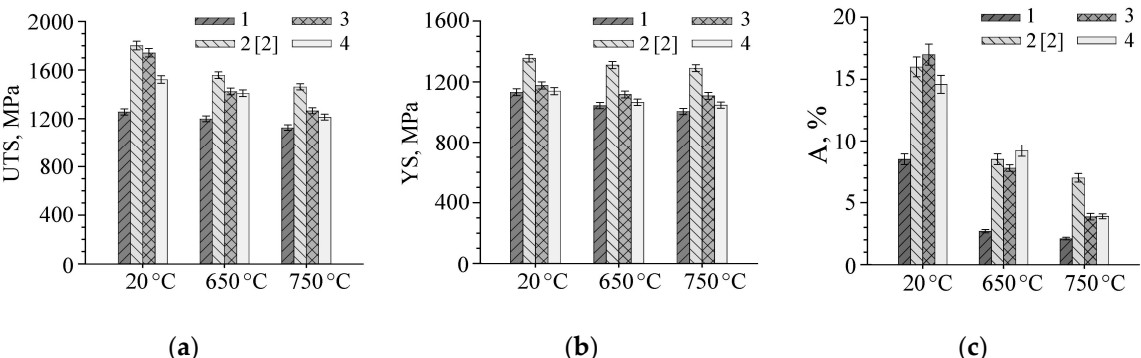

**Figure 5.** The tensile properties of the superalloy in conditions 1–4 obtained at 20, 650 and 750 °C: (**a**) ultimate tensile strength (UTS), (**b**) yield strength (YS), (**c**) elongation to rupture (A).

*3.4. Creep Properties*

Figure 6a–c shows the creep curves obtained for conditions 1–4 at 750 °C/700 MPa. One can see that the creep resistance significantly increased at transition from conditions 1 and 2 to conditions 3 and 4 (Figure 6a). Thus, the solid solution treatment provided a significant increment in the creep resistance. It should be ascribed to the small size of the $\gamma'$ precipitates, the small volume fraction of the primary $\gamma'$ phase, and the favorable $\gamma$ grain size.

The extended primary creep stage accompanying with a decrease of the creep rate was observed for condition 1 (Figure 6b,c). Note that a similar first stage was earlier observed in cast and single-crystal superalloys [10,12]. The extended primary creep stage can be associated with dislocation hardening, which can take place during creep of a coarse-grained or single-crystal superalloy. The sample in condition 1 was destroyed after 62 h until the third stage. In conditions 2–4, the first stage of creep was very short and the second stage of creep was accompanied by some oscillations in the creep rate (Figure 6b) that was reported earlier for forged superalloy in [7]. In conditions 2–4, the samples were not destroyed at 750 °C/700 MPa after 100 h. The solid solution treatment (conditions 3 and 4) provided higher creep resistance and a lower mean creep rate during the second stage as compared with conditions 1 and 2. Condition 4 showed slightly lower creep rate as compared with condition 3. However taking into account the tensile properties, condition 3 was chosen as the optimal one and, therefore, the creep tests at higher temperatures were performed for this condition.

Figure 6d–f represents the creep curves obtained for condition 3 at 750–850 °C/765–400 MPa. The creep curves obtained at 750–850 °C showed short first and secondary stages, followed by an increase of the creep rate corresponding to the third stage (Figure 6e,f). The creep rate at 750 °C/765 MPa was lower than those at 800–850 °C/550–400 MPa. Comparing creep curves obtained for condition 3 at

750 °C and different loadings, one can see that the minimal creep rate increased with increasing the loading from 700 to 765 MPa from $10^{-6}$ to $3 \times 10^{-6}$ s$^{-1}$ (Figure 6c,f). The following creep properties were achieved in the superalloy: at 650 °C/1200 MPa the creep rupture life and creep strain are 201 h and 5.4%; at 750 °C/765 MPa—124.6 h and 5.0%; at 800 °C/550 MPa—78.9 h and 22.1%; at 850 °C/400 MPa—38.6 h and 18.4%.

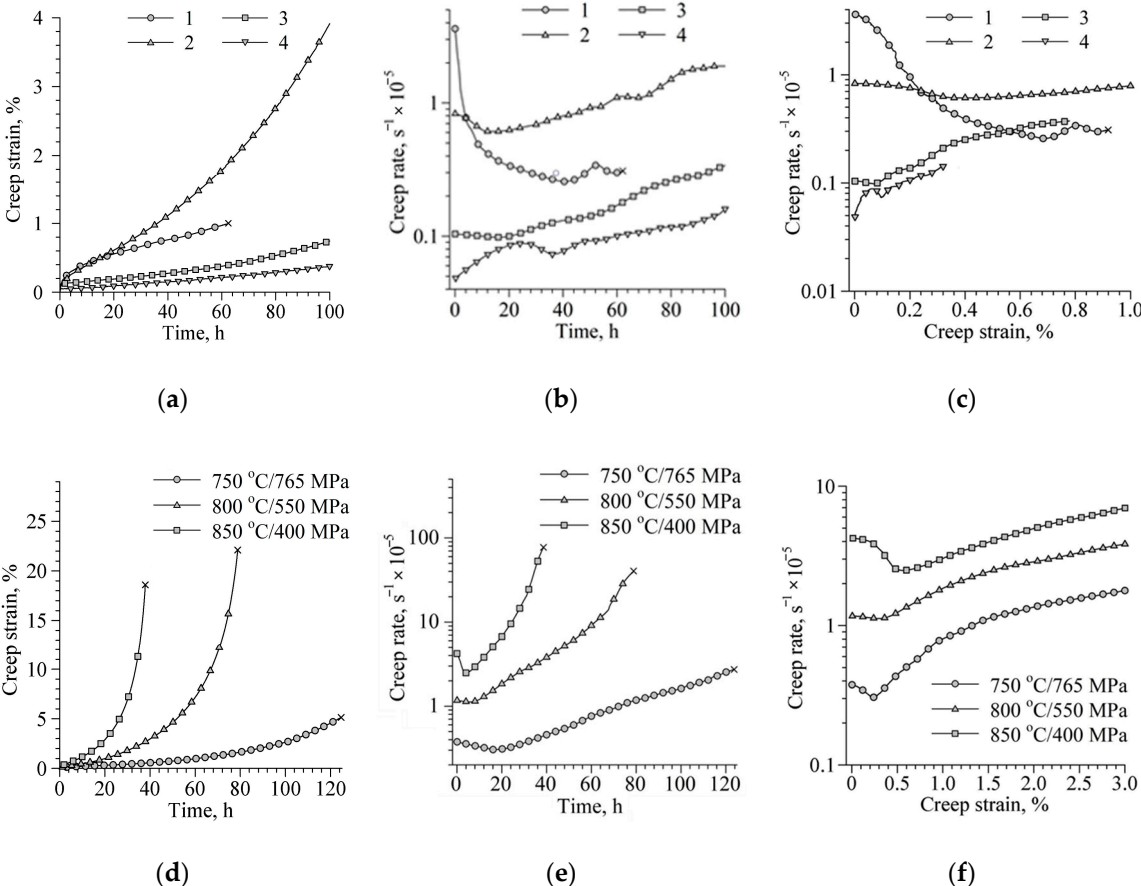

**Figure 6.** Creep behavior of the superalloy: (**a**) the creep curves obtained for conditions 1–4 at 750 °C/700 MPa, (**b**) the creep rate versus time plotted for conditions 1–4 at 750 °C/700 MPa, (**c**) the creep rate versus creep strain for conditions 1–4 at 750 °C/700 MPa, (**d**) the creep curves obtained for condition 3 at 750–850 °C/765–400 MPa, (**e**) the creep rate versus time plotted for condition 3 at 750–850 °C/765-400 MPa, and (**f**) the creep rate versus creep strain for condition 3 at 750–850 °C/765–400 MPa.

The obtained creep data were used to generate curves of stress versus LMP calculated by Equation (1). The resulting plots are shown in Figure 7, and a simple regression equation is included for estimating creep response at intermediate reply. The regression Larson–Miller equations of the superalloy are given from creep life ranging from 650 to 850 °C at different stresses (400–1200 MPa). Figure 7 indicates that the superalloy in conditions 3 and 4 has higher creep resistance in comparison with conditions 1 and 2. Particularly, creep resistance of the superalloy at 750–850 °C was appreciably higher in conditions 3 and 4 as compared with that in conditions 1 and 2. Comparing conditions 3 and 4, one can conclude that they showed similar creep resistance. Therefore, taking into consideration the tensile properties, *T* = 1170 °C can be assumed as the optimal solid solution treatment temperature for the SDZhS-15 superalloy.

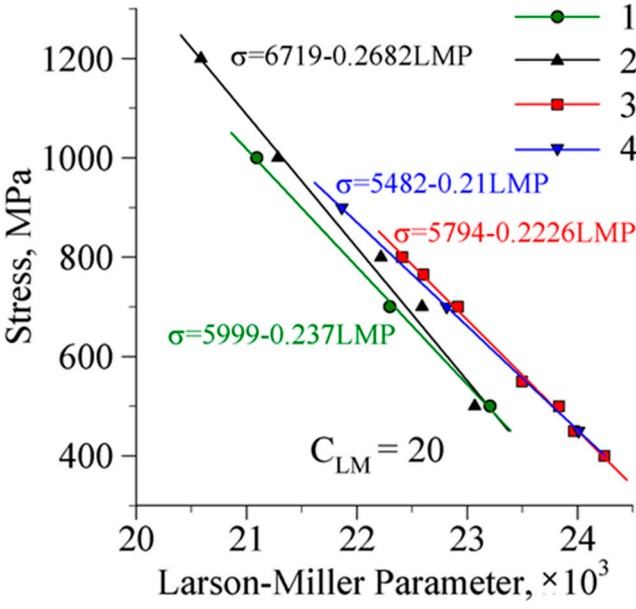

**Figure 7.** Larson-Miller parameter plotted for the superalloy in conditions 1–4.

### 3.5. Effect of Tensile and Creep Tests on The Microstructure and Fracture Behavior

Figure 8 represents the BSE images obtained near the fracture zone of the samples after creep testing at 850 °C. The creep strain at 850 °C did not lead to a significant change in the microstructure. It is worth noting that TCP phases were not detected. One can see that the creep strain led to crack nucleation along grain/interphase boundaries. This is probably associated with grain/interphase boundary sliding during creep strain and enhanced diffusivity along $\gamma$ grain and $\gamma/\gamma'$ interphase boundaries. In addition, coarse carbides and primary $\gamma'$ phase can also promote pore nucleation (Figure 8b,d).

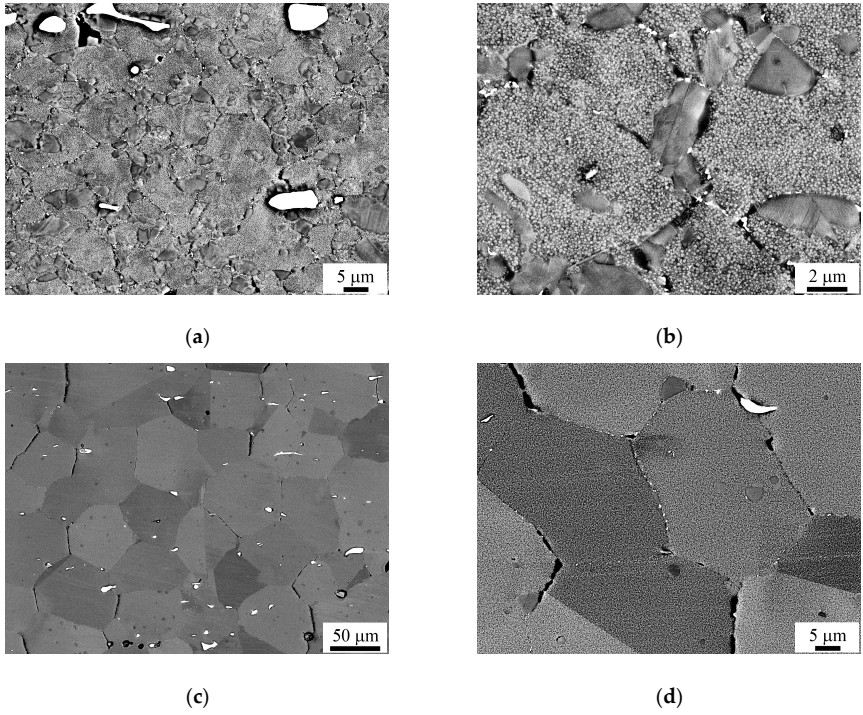

**Figure 8.** The BSE images obtained near the fracture zone of the samples after creep testing at 850 °C/450 MPa: (**a**,**b**) condition 3, (**c**,**d**) condition 4. The loading direction is horizontal.

## 4. Conclusions

Microstructure and mechanical properties of the newly designed Ni-based superalloy SDZhS-15 intended for disc applications were studied after different treatments. On this basis, effective processing, including hot forging with intermediate annealing, solid solution treatment, and ageing, was developed. Particularly, it was revealed that the post-forging solid solution treatment at temperatures higher than 1170 °C ($T_s$-50, where $T_s$ is the $\gamma'$ solvus temperature) led to a fast $\gamma$ grain growth that in turn led to a decrease of strength and ductility. The solution treatment at 1160–1170 °C (($T_s$-60)-($T_s$-50)) allowed to save fine-grained microstructure and to provide the formation of secondary $\gamma'$ precipitates with a size of around 0.1 μm.

Tensile and creep tests showed that the best combination of mechanical properties demonstrated condition 3 obtained via hot forging with intermediate annealing, solid solution treatment at 1170 °C, and ageing. The creep curves of the SDZhS-15 superalloy at 650–850 °C in the stress range of 400–1200 MPa were obtained. Under the creep at 650 °C/1200 MPa, the creep rupture life and strain of the superalloy were 201 h and 5.4%; at 750 °C/765 MPa—124.6 h and 5.0%; at 800 °C/550 MPa—78.9 h and 22.1%; at 850 °C/400 MPa—38.6 h and 18.4%. The creep resistance of the superalloy was compared in different conditions using the Larson–Miller parameter. Microstructure examination of the creep tested samples showed that a decrease in the creep resistance at 850 °C can be associated with enhanced diffusivity along $\gamma$ grain and $\gamma/\gamma'$ interphase boundaries leading to formation of cracks along the boundaries.

**Author Contributions:** Conceptualization, S.M. and V.I.; methodology, A.G., R.S., K.M., V.I., and S.M.; validation, A.G., V.I., and S.M.; formal analysis, V.I.; investigation, K.M., R.S., A.G., and N.P.; writing—original draft preparation, V.I.; writing—review and editing, S.M. and V.I.; visualization, A.G.; supervision, A.L.; project administration, V.I.; funding acquisition, S.M. All authors have read and agreed to the published version of the manuscript.

**Funding:** This research was supported by the Russian Science Foundation, Grant No. 18-19-00594.

**Acknowledgments:** The work was performed using the facilities of the shared services center "Structural and Physical-Mechanical Studies of Materials" at the Institute for Metals Superplasticity Problems of Russian Academy of Sciences.

**Conflicts of Interest:** The authors declare no conflict of interest.

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
