# Peer review of "Influence of Forging and Heat Treatment on the Microstructure and Mechanical Properties of a Heavily Alloyed Ingot-Metallurgy Nickel-Based Superalloy"

_metals, doi:10.3390/met10121606_

Round 1

Reviewer 1 Report

Dear authors,

The paper “Influence of Forging and Heat Treatment on the 2Microstructure and Mechanical Properties of a3Heavily Alloyed Ingot-Metallurgy Nickel-Based 4Superalloy” comprehensively shows that the process parameter strongly influences the microstructure and mechanical properties of the investigated alloy. The authors show this by a careful investigation carried out SEM and mechanical testing. At the end of the paper the failure mechanism are correlated to the microstructural. The paper is well organized. However, some revisions have to be done before publishing in Metals:

  • Line 146: The formation of carbides is at clearly higher temperatures in some other Ni-based alloys. Did you investigate the microstructure directly before and after the annealing heat treatment to be sure that the precipitation is within this heat treatment?
  • Line 146: please provide the volume fraction of the secondary gamma-prime phase, as this parameter is very important for the mechanical properties of the alloy.
  • Line 180: The partition behavior cannot be determined reliable in the fully annealed samples, as there is always a certain amount of gamma-prime in the measurement of the gamma phase. Please comment on this.
  • Line 188: Do you see different partition behavior in the different heat-treated conditions?
  • Line 200: Condition 3 and 4 show a very similar mechanical behavior at higher temperatures. Moreover, the strength of condition 1 is also close to condition 3 and 4. Please give the reason for this behavior. What is the main hardening mechanism at these temperatures?
  • Line 208: The symbols for condition 4 are not correct in fig. 6a.
  • Line 208: Condition 4 shows the lowest creep rate at 750 °C / 700 MPa. However, their LMP is lower than the one of condition 3. Is there a mistake?
  • Line 216: Please describe the reason for the extended first stage during creep in condition 1.
  • Line 251: Do you see a growth of the primary precipitates after creep tests?

Author Response

Dear Editor, Dear Reviewer,

Thank you for reviewing our manuscript Metals-1022156. Concerning the comments we can give the following explanations.

Reviewer #1: The paper “Influence of Forging and Heat Treatment on the Microstructure and Mechanical Properties of a Heavily Alloyed Ingot-Metallurgy Nickel-Based Superalloy” comprehensively shows that the process parameter strongly influences the microstructure and mechanical properties of the investigated alloy. The authors show this by a careful investigation carried out SEM and mechanical testing. At the end of the paper the failure mechanism are correlated to the microstructural. The paper is well organized. However, some revisions have to be done before publishing in Metals:

Comment (Reviewer #1). Line 146: The formation of carbides is at clearly higher temperatures in some other Ni-based alloys. Did you investigate the microstructure directly before and after the annealing heat treatment to be sure that the precipitation is within this heat treatment?

We investigated the microstructure both before ageing and after ageing. One additional figure (Fig. 2a) corresponding to the condition before the ageing treatment has been added. Before ageing only relatively coarse carbides precipitated at higher temperatures were observed. The ageing treatment resulted in formation of small carbides having a size of less than 1 mkm as is seen in Fig. 2c,d. Some corrections and additions have been made in the caption and the text describing the microstructure conditions.

Comment (Reviewer #1). Line 146: please provide the volume fraction of the secondary gamma-prime phase, as this parameter is very important for the mechanical properties of the alloy.

Information about the volume fractions of the secondary gamma prime phase has been added to the text.

Comment (Reviewer #1). Line 180: The partition behavior cannot be determined reliable in the fully annealed samples, as there is always a certain amount of gamma-prime in the measurement of the gamma phase. Please comment on this.

Energy-dispersive X-ray spectroscopic (EDS) analysis was carried out for the cast condition subjected to homogenization annealing followed by slow cooling with a rate of 25°C/h. The slow cooling led to coarsening of the gamma prime phase and significant reducing the fraction of the dispersed gamma prime precipitates. That allowed us to evaluate the partitioning behavior of the alloying elements taking into account gamma grain areas free of the gamma prime precipitates versus coarse gamma prime particles. Some explanations and corrections have been made in the methods and section 3.2.

Comment (Reviewer #1). Line 188: Do you see different partition behavior in the different heat-treated conditions?

The partitioning behavior of the alloying elements was investigated only for the cast condition subjected to homogenization annealing followed by slow cooling. In the other conditions containing a high volume fraction of the gamma prime precipitates within the gamma grains it does not make sense because, as mentioned by the Reviewer in the previous comment, a certain amount of the gamma prime phase will be always presented in gamma grains.

Comment (Reviewer #1). Line 200: Condition 3 and 4 show a very similar mechanical behavior at higher temperatures. Moreover, the strength of condition 1 is also close to condition 3 and 4. Please give the reason for this behavior. What is the main hardening mechanism at these temperatures?

Discussion concerning tensile properties was given in more details. The following text has been added to section 3.3:

The decrease of the strength properties (UTS and YS) in conditions 3 and 4 in contrast to those in condition 2 is associated with coarser gamma grain size in conditions 3 and 4. Conditions 3 and 4 showed similar tensile properties at 650 and 750°C, although the gamma grain size in condition 4 was coarser than that in condition 3 (60 mm vs. 16 mm). Apparently, the higher volume fraction of the secondary gamma prime precipitates in condition 4 in contrast to condition 3 (63.5% vs. 58%) promoted higher strength that compensated for the loss of strength due to larger gamma grain size in condition 4. Comparing conditions 1, 3 and 4, one can see that the yield strength values in these conditions were similar (especially in conditions 1 and 4). This suggests that the yield strength value weakly depends on the gamma grain size in the range of d~20-200 mm and mostly depends on the volume fraction of the secondary gamma prime precipitates. Note that the higher volume fraction of the secondary gamma prime precipitates in conditions 1 and 4 as compared with that in condition 3 (68 and 63.5% vs. 58%) apparently compensated for the loss of the yield strength due to larger gamma grain size in conditions 1 and 4. At the same time, the UTS values and the elongations in conditions 3 and 4 were found to be appreciably higher than those in condition 1. This should be contributed to finer gamma grain size in conditions 3 and 4, which promoted a uniform development of plastic deformation that in turn promoted attaining higher elongations and UTS values.

Comment (Reviewer #1). Line 208: The symbols for condition 4 are not correct in fig. 6a.

The symbols for condition 4 in fig. 6a have been corrected.

Comment (Reviewer #1). Line 208: Condition 4 shows the lowest creep rate at 750 °C / 700 MPa. However, their LMP is lower than the one of condition 3. Is there a mistake?

Unfortunately, we did some mistakes. Figure 7 has been redrawn. In the new version of figure 7, the LMP for conditions 3 and 4 are very similar.

Comment (Reviewer #1). Line 216: Please describe the reason for the extended first stage during creep in condition 1.

Extended primary creep stage with reduced creep rate is typically observed in cast and single-crystal superalloys. Note that condition 1 is the coarse-grained cast condition subjected to heat treatment. The extended primary creep stage can be associated with dislocation hardening, which can take place during creep of a coarse-grained or single-crystal superalloy. One sentence has been added to the text.

Comment (Reviewer #1). Line 251: Do you see a growth of the primary precipitates after creep tests?

We did not detect the growth of the primary precipitates after creep tests.

Thus, in accordance with the Reviewer’s comments, the methods, results and conclusions have been revised. In addition, we have excluded some inaccuracies and improved the English throughout the manuscript. In the marked version of the revised manuscript the added text is marked by color.

Once again many thanks for reviewing the manuscript.

Best regards,

Shamil Mukhtarov

Reviewer 2 Report

The manuscript "Influence of Forging and Heat Treatment on the Microstructure and Mechanical Properties of a Heavily Alloyed Ingot-Metallurgy Nickel-Based Superalloy" has been reviewed.

The manuscript is clear, balanced and well-organized.

However some minor changes are required:

Line 73: workpieces not worpieces;

Lines 104-105: the reason why angles less than 2° were excluded from the consideration should be motivated;

Line 110: it should be explained the standard (ISO 6892, ASTM E8?) followed in the selection of the strain rate (8.3 x 10-4 s-1);

Fig. 5 (picture and caption): elongation is usually indicated as A% (not delta);

Fig. 5: it is not clear why elongation is considerably reduced with increasing test temperature. Please explain!

Line 256: treatments not treatment.

Author Response

Dear Editor, Dear Reviewer,

Thank you for reviewing our manuscript Metals-1022156. Concerning the comments we can give the following explanations.

Reviewer #2: The manuscript is clear, balanced and well-organized. However some minor changes are required:

Comment (Reviewer #2). Line 73: workpieces not worpieces.

The word “workpieces” has been corrected.

Comment (Reviewer #2). Lines 104-105: the reason why angles less than 2° were excluded from the consideration should be motivated.

In accordance with F.J. Humphreys (Journal of Microscopy (1999) V. 195, p. 170-185), “the accuracy of determination of the absolute orientation is typically ~1° and depends upon calibration and sample alignment”. Therefore, the misorientations less than 2° were not taken into consideration. Note that most of the investigators using EBSD analysis make the same (see for example also https://link.springer.com/article/10.1134/1.1992602). We have added the mentioned reference to the methods. The references have been renumbered.

Comment (Reviewer #2). Line 110: it should be explained the standard (ISO 6892, ASTM E8?) followed in the selection of the strain rate (8.3 × 10-4 s-1).

In accordance with ASTM E8 the speed of the testing machine should be between 0.05 and 0.5 mm/min. According to ISO 6892 the strain rate should be not lower then 8×10-3 s-1. In our tensile tests both these conditions were met. The following sentence has been added to the methods: “A constant speed of the testing machine was 0.5 mm/min that corresponded to the initial strain rate of έ=8.3×10-4 s-1.

Comment (Reviewer #2). Fig. 5 (picture and caption): elongation is usually indicated as A% (not delta).

“δ, %” has been replaced by “A, %”.

Comment (Reviewer #2). Fig. 5: it is not clear why elongation is considerably reduced with increasing test temperature. Please explain!

It is an open issue. However it should be noted that the decrease of ductility with increasing the test temperature was also observed in other superalloys (see for instance 1. Zheng L., Schmitz G., Meng Y. et. al. Critical Reviews in Solid State and Materials Sciences (2012) 37, p. 181-214; 2. Rao G.A., Kumar M., Srinivas M. Materials Science and Engineering A355 (2003) p. 114-125; 3. Braun A.R., Radavich J.F. Superalloys 718 (1989) p. 623-629.)

Comment (Reviewer #2). Line 256: treatments not treatment.

The word “treatments” has been corrected.

Thus, in accordance with the Reviewer’s comments, the methods, results and conclusions have been revised. In addition, we have excluded some inaccuracies and improved the English throughout the manuscript. In the marked version of the revised manuscript the added text is marked by color.

Once again many thanks for reviewing the manuscript.

Best regards,

Shamil Mukhtarov

Round 2

Reviewer 1 Report

Dear authors, the paper has clearly been improved and is now suitable to be published.